# Estimating 3D Chlorophyll Content Distribution of Trees Using an Image Fusion Method Between 2D Camera and 3D Portable Scanning Lidar

**Fumiki Hosoi \*, Sho Umeyama and Kuangting Kuo**

Graduate School of Agricultural and Life Sciences, The University of Tokyo, Yayoi 1-1-1, Bunkyo-ku, Tokyo 113-8657, Japan; umeyama-sho@g.ecc.u-tokyo.ac.jp (S.U.); leo031350727@g.ecc.u-tokyo.ac.jp (K.K.)
\* Correspondence: ahosoi@mail.ecc.u-tokyo.ac.jp

**Abstract:** An image fusion method has been proposed for plant images taken using a two-dimensional (2D) camera and three-dimensional (3D) portable lidar for obtaining a 3D distribution of physiological and biochemical plant properties. In this method, a 2D multispectral camera with five bands (475–840 nm) and a 3D high-resolution portable scanning lidar were applied to three sets of sample trees. After producing vegetation index (VI) images from multispectral images, 3D point cloud lidar data were projected onto the 2D plane based on perspective projection, keeping the depth information of each of the lidar points. The VI images were 2D registered to the lidar projected image based on the projective transformation and VI 3D point cloud images were reconstructed based on the depth information. Based on the relationship between the VI values and chlorophyll contents taken by a soil and plant analysis development (SPAD)-502 plus chlorophyll meter, 3D distribution images of the chlorophyll contents were produced. Similarly, a thermal 3D image for a sample was also produced. The resultant chlorophyll distribution images offered vertical and horizontal distributions, and those for each orientation for each sample, showing the spatial variability of the distribution and the difference between the samples.

**Keywords:** chlorophyll distribution; image fusion; lidar; multispectral camera; plant physiology; plant biochemistry; 3D plant property; 3D imaging

## 1. Introduction

Accurate measurements of plant physiological and biochemical properties are necessary to understand plant functions. Various imaging techniques have been proposed for the measurements, which include the estimation of plant pigments, such as chlorophyll or carotenoids using spectral images [1–3], stomatal conductance, transpiration and water potential using a thermal camera [4–6], and photosynthetic ability based on chlorophyll fluorescence imaging [7,8]. Such imaging techniques are mainly two-dimensional (2D). However, owing to the three-dimensional (3D) attributes of the plant structure, plant organs distribute three-dimensionally, so that the physiological and biochemical properties also have a 3D distribution. Thus, the above-mentioned 2D imaging should be upgraded to 3D imaging and a method that allows an integrated analysis of 3D structural, and physiological and biochemical properties should be developed for a better understanding of plant functions. Portable scanning lidars (also called terrestrial laser scanners) have been previously used to obtain the 3D structural properties of plants because of their advantageous attributes, such as portability, efficiency of data collection, high ranging accuracy, and spatial resolution. Various plant structural properties have been extracted using portable scanning lidars, e.g., tree height, biomass, leaf area density, leaf inclination angle, branching pattern, etc. [9–15]. Although structural information has been extracted

using lidars, their ability to provide a detailed 3D reconstruction of plants can also be utilized to obtain 3D distributions of physiological and biochemical plant properties. Such applications have been reported in several studies. A portable scanning lidar that can provide a colored point cloud was used for estimating the chlorophyll contents within trees [16]. Chlorophyll (abbreviated as Chl) and nitrogen contents were also estimated by a portable scanning lidar with a green laser source [17,18]. These methods can provide chlorophyll contents, which relate to biochemical processes, associated with the structural properties. However, only visible bands can be used in these methods, and for this reason, the available physiological and biochemical information is limited. A solution to this limitation is to use a multispectral or hyperspectral lidar. The optical information of plants obtained from these devices provides useful insights into the physiological and biochemical properties, such as Chl and nitrogen contents, species classification, leaf water content [19–23]. Although these are very useful tools for studying plant properties, only optical information is obtained. In addition, the instruments are not commercially available and thus it is difficult for them to be widely used for plant measurements. Another direction is the fusion between the lidar and 2D images. Portable lidar images and 2D thermal, spectral, and Chl fluorescence images were fused in small plants and 3D plant responses to herbicide treatments were clearly shown [24,25]. The fusion of an RGB camera and depth camera (one of the lidar instruments) was used for soybean canopy analysis [26]. The image fusion was conducted by a portable lidar and thermal camera to obtain plant spatial temperature distribution [27]. As shown in these cases, image fusion techniques allow the combination of lidars and various 2D imaging cameras, in addition to the optical cameras. Thus, the application of the fusion technique has the potential to increase the information available on the 3D distributions of physiological and biochemical properties. However, detailed information about the 3D distribution of plant physiological biochemical properties was not fully extracted in the aforementioned studies. This implies that the potential of the fusion technique in plant measurements has not been fully demonstrated as yet. In the methods adopted by [24,25], the texture mapping technique [28,29] was applied for fusion, in which a lidar-derived 3D point cloud image was converted to a polygonal surface mesh image. It is usually difficult for the lidar-derived 3D point cloud image of a plant to be converted to a polygonal surface mesh image because the 3D point distribution of a leaf canopy is very complicated. Therefore, the method is applicable only to small plants with a simple structure. To apply the fusion method to various types and sizes of plants, a method is required in which the 2D image is directly fused to the 3D point cloud image without conversion to the polygonal surface mesh. In the studies by [16,26], fusion systems were designed and assembled for specific 2D cameras and lidars. To effectively obtain plant physiological and biochemical information together with the structural features, the fusion method should be easily applicable to different types of 2D cameras and lidars.

In this study, we propose a fusion method between the 2D camera and 3D portable lidar images, suitable for extracting plant structural and physiological information. In this method, each pixel in a 2D camera image of a plant is directly associated with the corresponding lidar point without polygonal surface mesh conversion. It is further shown that the composite 3D images of plants obtained by this method can be used to derive 3D distributions of plant biochemical properties. The applicability of the present fusion method to different 2D cameras and lidars in the plant measurements is also demonstrated. As an example of this demonstration, derivation of 3D distribution of chlorophyll contents was attempted with a multispectral camera and portable scanning lidar.

## 2. Materials and Methods

Two study sites were chosen for this study. The first site was the Shinjuku Gyoen National Garden in the center of Tokyo (35°59′N, 140°02′E), where about 250 species and 2000 trees grow. From this site, a single Yoshino cherry (*Prunus × yedoensis*) was chosen as a specimen of a deciduous tree (herein called sample A). The height of the cherry tree was 12.5 m. The north side of sample A was closed by other trees and the south, west, and east sides were open space. The measurement date was October 30, 2018, when autumnal tints of sample A had already started. The condition of the sky was clear. The second study site was a mixed plantation in Ibaraki Prefecture, 40 km northeast of central Metropolitan Tokyo, Japan (35°59′N, 140°02′E), whose dominant tree species were Japanese cedar (*Cryptomeria japonica* [L.f.] D. Don), Japanese red pine (*Pinus densiflora* Siebold & Zuccarini), ginkgo (*Ginkgo biloba* Linnaeus), and Japanese zelkova (*Zelkova serrata* [Thunberg] Makino). From this site, a single bamboo leaf oak tree (*Quercus myrsinaefolia* Blume) and the community of the bamboo leaf oak trees were chosen as specimens of evergreen trees (herein called samples B and C, respectively). The height of sample B was 6.5 m, and the south and southwest sides were open while the other sides were enclosed by other trees. In the case of sample C, the height was 17.1 m and the southwest side was open while the other sides were enclosed by other trees. The measurement date was October 10, 2018. The condition of the sky was clear.

Laser scans of the samples were conducted using a portable scanning lidar (Focus 3D X330; FARO Technologies Inc., Lake Mary, FL, USA). The lidar calculates distances by the phase shift method and has a range resolution of 3 mm at a measurement range of 20 m with the measurable range of 0.6 to 330 m and range accuracy of ±2 mm. Three equidistant measurement positions for the lidar scan were set around each sample at about 25 m, 6.5 m, and 25 m for samples A, B, and C, respectively. The distance determines the laser footprint size and also the horizontal resolution of the lidar image. Each position was ~120° apart from each other toward the azimuth direction to capture the whole canopy. In the case of sample C, a part of the canopy, whose area was 24.0 × 9.1 × 17.1 m, was chosen as the target region for the laser scan. Spectral images were taken using a multispectral camera (Red Edge, Micasense Inc., Seattle, WA, USA), which could simultaneously take five images with the bands of blue (475 nm), green (560 nm), red (668 nm), red edge (717 nm), and near-infrared (840 nm). The resolution of the image was 1280 × 960. In the present fusion method, the camera positions were determined to obtain similar projected images of the camera and lidar (described later). For this purpose, the camera was put on the points on the lines connecting the sample center and lidar positions, and the distance between the camera position and sample center was adjusted (Figure 1). The camera positions were set to the same positions as those of the lidar when the whole target canopy was included in the image at the positions (this was the case for sample B). When the camera could not cover the whole canopy in the lidar positions owing to an insufficient angle of view or too large sample size, the distance between the camera position and sample center was made larger than the distance between the lidar position and the center, to cover the whole canopy (Figure 1: this was applied to samples A and C). In this case, the sample center, the lidar position, and the center of the camera monitor were aligned on a straight line by viewing the camera monitor and moving the camera body. A reference board with 18% reflectance was put on the ground near the samples and was photographed together with the samples. This board was used for converting pixel values of the multispectral images to reflectance.

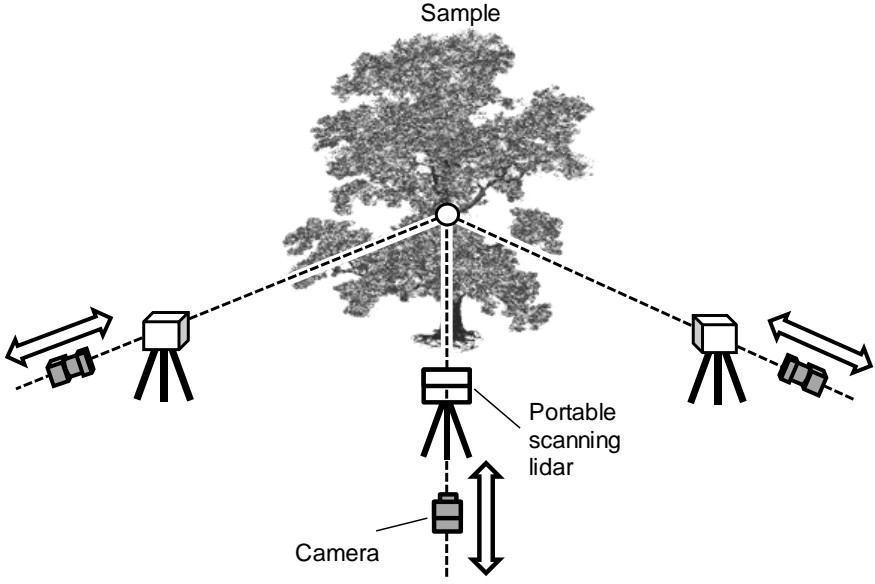

**Figure 1.** Schematic illustration of the positional relation of a camera and portable scanning lidar.

For fusion between the multispectral and lidar images, 3D point cloud lidar data was first projected onto the 2D plane based on perspective projection (see Figure 2) using the equations below.

$$x' = d\frac{x}{z}, \ y' = d\frac{y}{z}$$

(1)

where $(x, y, z)$ and $(x', y')$ are the coordinates of a point in the 3D point cloud image and on the projection plane, respectively. $d$ is the distance between the lidar position and projected plane, which determines the scale of the projected target image within the projection plane. $d$ is adjusted as the whole canopy is included in the projection plane. Each point in the 3D image is converted to each pixel in the 2D image (called lidar projected image). The z-value of each 3D point (depth information) was kept as an attribute value of each pixel in the projected image. This projected image was similar to the multispectral image because the latter were taken at the same position as the lidar position or the position on the line connecting the sample center and the lidar position. That is, the projection plane for the 3D image was parallel to the image plane on the multispectral camera. Then, vegetation index (VI) images, herein normalized difference vegetation index (NDVI) or green normalized difference vegetation index (GNDVI) [30] images, were produced from the spectral images. Five corresponding points between the VI and lidar projected images were chosen manually by eye, which had distinguishable parts such as canopy edges or branches. Using the corresponding points, the VI image was 2D registered to the lidar projected image based on projective transformation, where the transformation matrix was estimated by a least-squares method based on the corresponding points. The VI value of each point was added as an additional attribute value in each of the pixels in the lidar projected image, and a composite 2D image between VI and the lidar projected image was produced. Then, the composite image was re-constructed as the VI 3D point cloud image based on the z-value included as an attribute value in each pixel. These processes were repeated for all lidar images taken at different positions (three images for each sample) and the reconstructed images were co-registered and merged, providing VI 3D images of the samples. Although such co-registration between the lidar projected image and the 2D image taken by the camera has been reported in the other applications, e.g., terrain mapping [31], the method has not been tried and validated to the measurements of plants with a complicated structure, and the positional relationship of the camera and lidar has not been considered before. The composite image obtained above included both leaves under sunlight and in the shade. Since the reference panel was put under sunlight, the reflectance of the leaves in the shade was not a correct value. Consequently, the points corresponding to leaves in the shade were distinguished by determining the threshold value to

judge the shade based on the VI values of shaded leaves in the image. These points in the shade were not used for subsequent Chl content estimation.

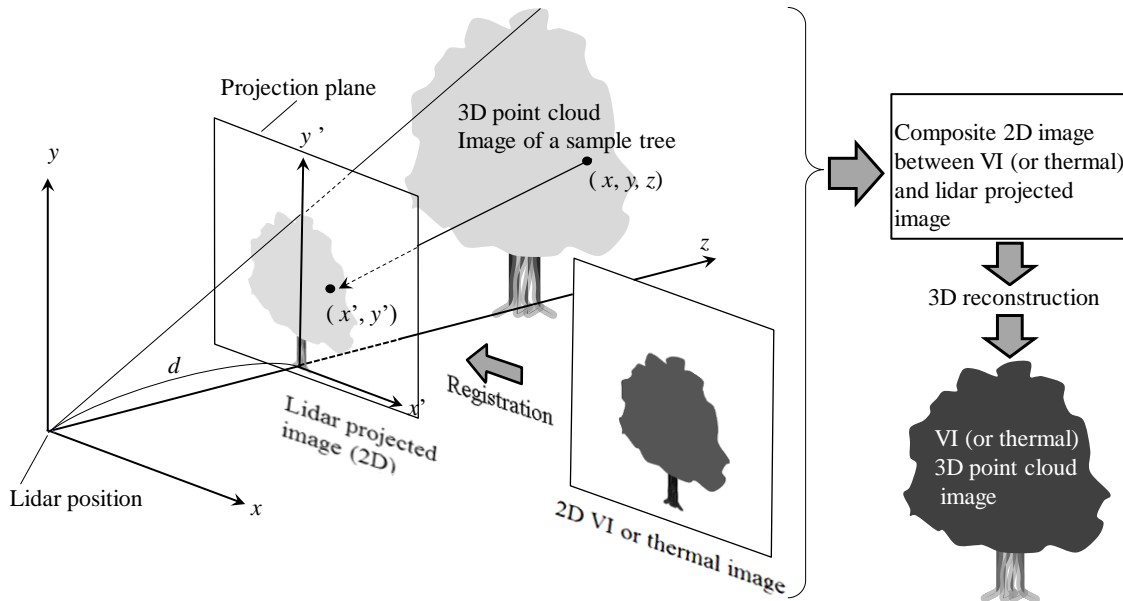

**Figure 2.** Image fusion between the 2D camera and 3D lidar-derived point cloud images of a sample tree. $(x, y, z)$ and $(x', y')$ are coordinates of a point in the 3D point cloud image and on the projection plane, respectively. $d$ is distance between the lidar position and the projected plane.

SPAD (Soil and Plant Analysis Development)-502 Plus chlorophyll meter (Konica Minolta Sensing, Osaka, Japan) was used to estimate Chl contents, in which the difference between the absorbance of leaves at 650 nm and 940 nm was used for the estimation. Several previous studies report that the SPAD values correlate well to the actual Chl content and thus can be used as a substitute [32–37]. SPAD values can be converted into the actual Chl values using the correlation equation, as needed. Leaves were randomly selected from samples and they were marked by white markers. The numbers of the leaves were 20, 19 and 29 for sample A, B, and C, respectively. The marked leaves were photographed by the multispectral camera and measured by the SPAD. The VI values were derived from the marked leaf images, and the relationships between the VI and SPAD values corresponding to each marked leaf were examined. In the case of sample A, exponential curve fitting was conducted due to NDVI saturation for SPAD values. Linear fitting was conducted for the other samples due to no saturation of GNDVI values. From these relationships, conversion equations of the VI to SPAD values were obtained. Based on the equations, pixel values in VI images were converted to the SPAD values and a 3D distribution image of the Chl contents was produced for each sample. Moreover, for demonstrating the applicability of the present fusion method using 2D cameras other than the multispectral one, thermal images were taken with a thermal camera (PI 640; Optris GmbH, Berlin, Germany) were also combined with the lidar images of sample C. The image size of the camera was 640 × 480 pixels with a field of view of 33° × 25° and the spectral range was 7.5 to 13.0 μm and the accuracy of the temperature was ± 2 °C or ± 2%, with the measurable range of −20 to 100 °C. The emissivity was set to 0.99. The measurement date in the case of the thermal camera and lidar experiment was December 2018. The measurement positions and the fusion process were similar to the case of the multispectral camera used for sample C, as explained above.

The 3D spatial analysis was conducted based on the Chl distribution images. To estimate the vertical distributions, the Chl images were divided into a horizontal layer with thicknesses of 1.0 m, 0.5 m, and 1.0 m for samples A, B, and C, respectively. The Chl value of each horizontal layer was calculated by averaging the Chl values of all points within a layer. To obtain the Chl horizontal

distributions within each horizontal layer, each layer was divided into numerous cubic cells that were regularly 3D arranged with no space in between. The size of the cell was 1.0 m × 1.0 m × 1.0 m for samples A and C and 0.5m × 0.5m × 0.5m for sample B. The Chl value of a certain horizontal position within a horizontal layer was obtained by averaging the Chl values of all points included in a cell. By repeating the process for all cells, the Chl horizontal distributions in each horizontal layer were estimated. To obtain the Chl distributions in each orientation, the coordinates of each point within the Chl image were converted into polar coordinates with which the origins were determined from the trunk centers of samples A and B, and from the center of the 3D Chl image in the case of sample C. The points within the Chl image were allocated into east, west, south, and north regions based on the azimuth angle of each point (east: 45–135°, west: 225–315°, south: 135–225°, north: 45–315°). The vertical Chl distributions were calculated in each of the regions.

## 3. Results

Figure 3 illustrates the relationships between the VI and SPAD values for each of the samples; the VI and SPAD values have a good correlation with $R^2$ values from 0.818 to 0.936. Leave one out cross-validation was applied to the correlations and the root mean square error (RMSE) values were 3.58, 2.29, and 4.48 in samples A, B, and C, respectively, implying an accurate estimation of Chl contents. Figure 4 shows the 3D Chl images produced by the present image fusion method. In these images, the 3D distribution of Chl contents can be observed freely from different view angles and any region can be chosen for 3D spatial analysis. In the comparison of Chl contents among the samples, the values were lower for sample A than those of the other two samples, possibly because of the influence of autumnal tints. It was evident that the Chl contents were unevenly distributed through the whole canopy in all samples. In particular, the variation was larger for sample C. As shown in Figure 4d, the present fusion method could also be applied to the thermal image. The temperature of the canopy ranged from 3.3 to 6.6 °C and was different in each part of the canopy, depending mainly on the incident light intensity. The registration errors of the present image fusion are shown in Table 1.

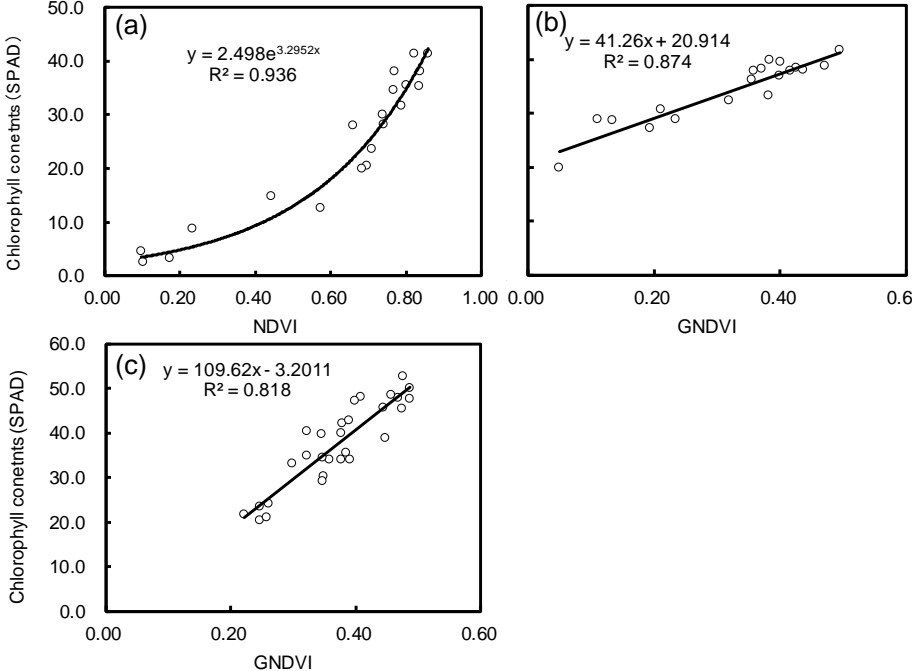

**Figure 3.** Relationships between VI (Vegetation Index) and SPAD values in each of the samples. (**a**) Sample A, (**b**) Sample B, and (**c**) Sample C. NDVI: Normalized difference vegetation index, GNDVI: Green normalized difference vegetation index.

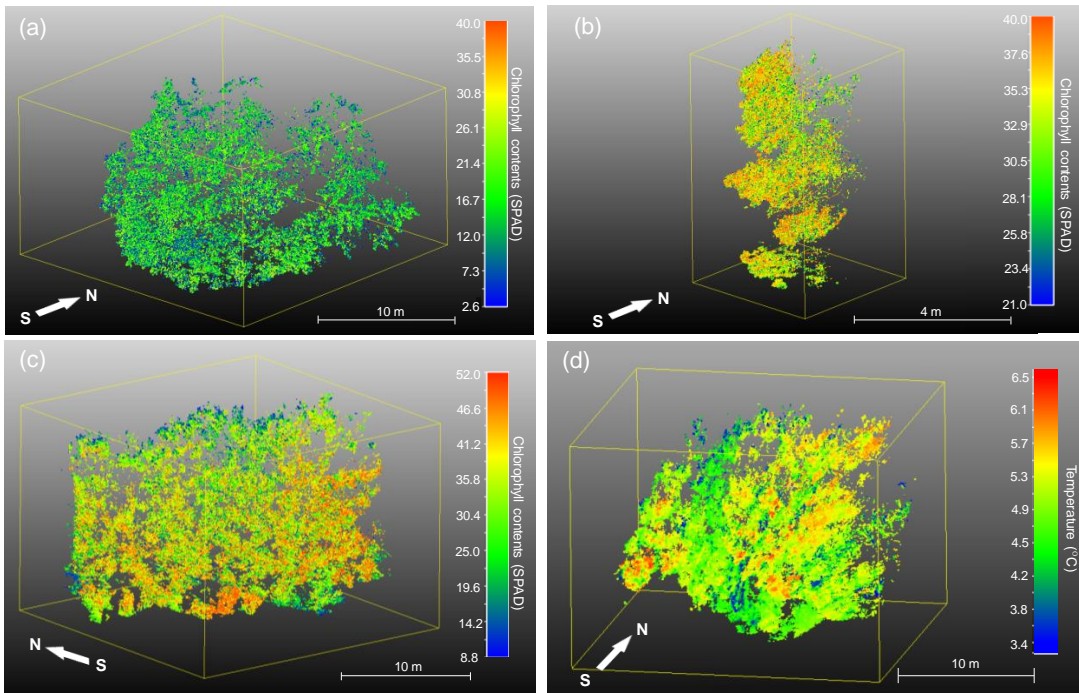

**Figure 4.** 3D chlorophyll and thermal point cloud images of the samples based on the image fusion between a multispectral camera or thermal camera and a portable scanning lidar. (**a–c**) are the 3D chlorophyll images of sample A, B, and C, respectively, and (**d**) is a 3D thermal image of sample C.

**Table 1.** Registration errors of the present image fusion method between a 2D multispectral camera or thermal camera images and a 2D projected lidar image. Values in parentheses are errors in the case of the thermal camera and the other values are errors in the case of the multispectral camera.

| Sample | Registration Error | | | | | |
| --- | --- | --- | --- | --- | --- | --- |
| | Pixel | | | Actual Dimension (m) | | |
| | Mean | Max. | Min. | Mean | Max. | Min. |
| A | 1.6 | 5.1 | 0.0 | 0.06 | 0.19 | 0.00 |
| B | 1.8 | 3.2 | 0.0 | 0.02 | 0.03 | 0.00 |
| C | 4.3 (4.8) | 12.0 (8.1) | 1.0 (2.2) | 0.16 (0.18) | 0.46 (0.30) | 0.04 (0.08) |

The errors were calculated for 15 corresponding points in each sample. The mean errors were 1.6 to 4.3 pixels on average in the case of the multispectral camera and 4.8 in the case of the thermal camera. These mean errors corresponded to 0.02 m to 0.16 m for the multispectral camera and 0.18 m for the thermal camera in actual dimensions, and they were accurate enough for spatial analysis using the present cell sizes of 0.5 and 1.0 m. Figure 5 shows the vertical distributions of Chl contents for each sample. In the case of sample A, the values decreased as the height increased. While the values also tended to be higher in the layers at a height of 1.0 to 3.0 m in the case of sample B, it was lowest at 4.0 m height. In the vertical profile of sample C, the Chl contents decreased as the height increased.

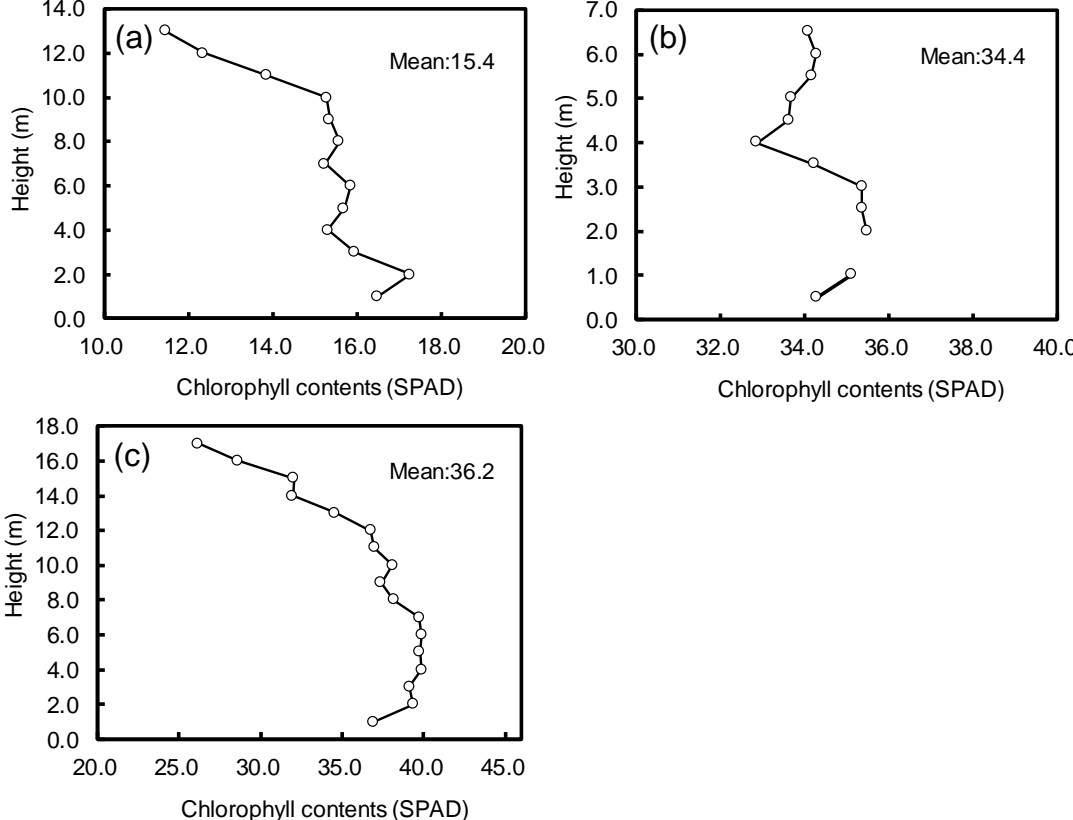

**Figure 5.** Vertical distributions of chlorophyll (Chl) contents for each sample. (**a**) Sample A, (**b**) Sample B, and (**c**) Sample C. The absence of a line in the profile between 1.0 to 2.0 m height for sample B means that there were no leaves at the height of 1.5 m.

The change of the values in sample C was larger than that in the other two samples. Figure 6 shows the horizontal distribution of Chl contents in each horizontal layer. It is evident that the Chl contents within the canopy were distributed unevenly with dispersion. It can also be observed that the layers with high Chl values in Figure 5 include more cells with higher Chl values. Figure 7 shows the vertical distributions of the Chl contents of each region facing each orientation for each sample. Although the profiles for samples A and C were different in each orientation, the overall tendency was that the Chl contents were higher in the lower layer and vice versa. Although the profile of the south side in sample B was similar to that of the whole canopy in Figure 6, the profiles were considerably different among each orientation.

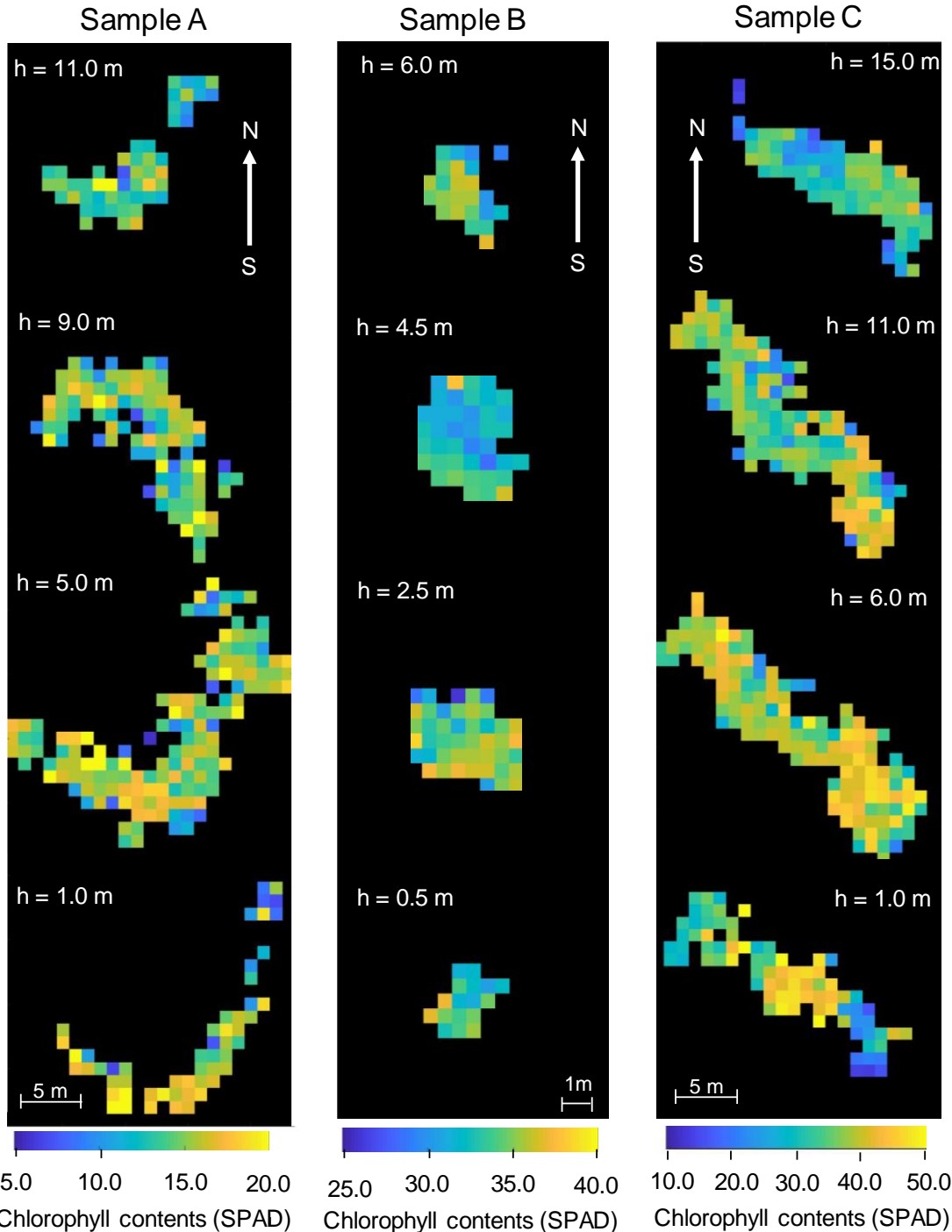

**Figure 6.** Horizontal distribution of Chl contents in each horizontal layer. h = height of the layer.

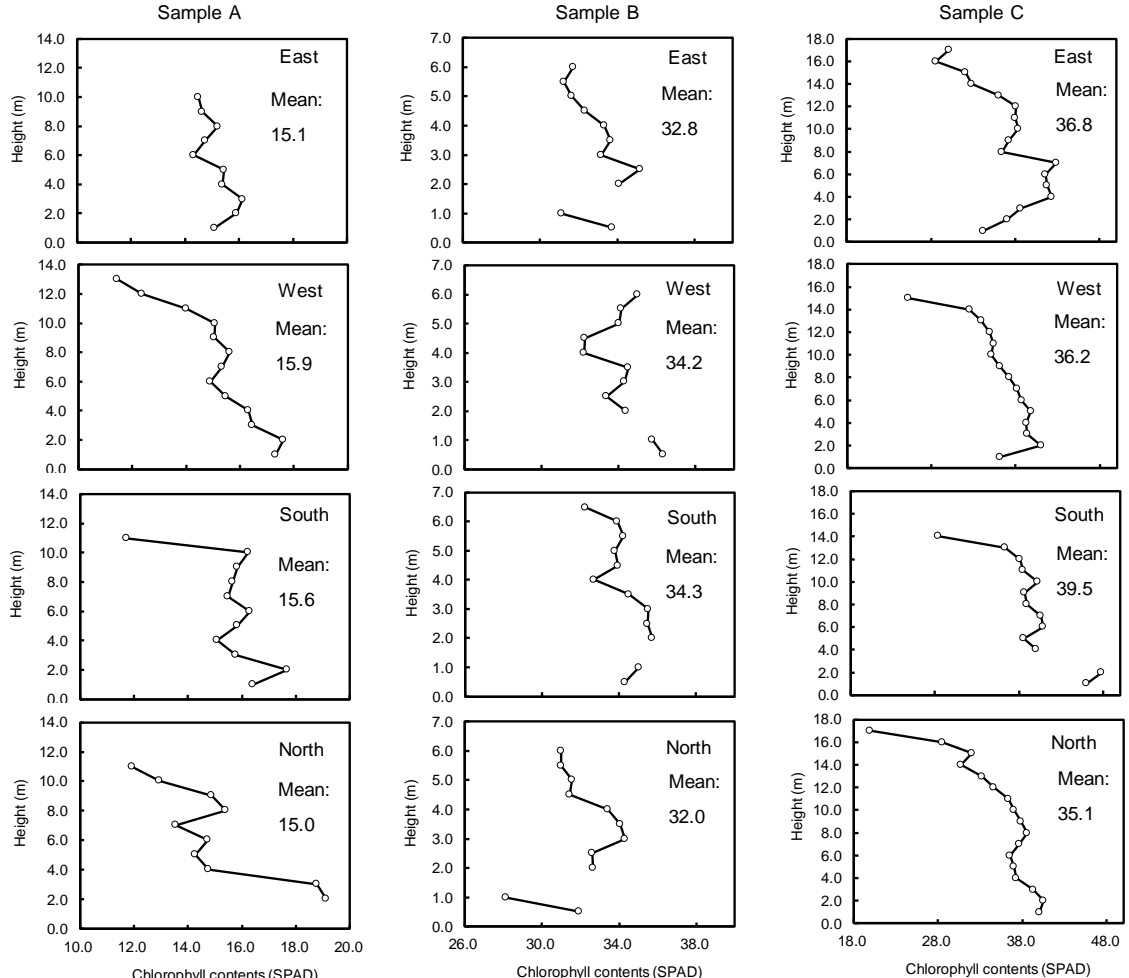

**Figure 7.** Vertical distributions of Chl contents of each region facing each orientation for each sample. The absence of a line in the profiles of samples B and C means that there were no leaves at those heights.

## 4. Discussion

There are several possible factors that affect the distribution of leaf pigments, e.g., light environment [38,39], leaf senescence [40], tree age [41], nitrogen and water availability [42–44], etc. Effective methods for measurements of the 3D spatial distribution of leaf pigments are needed for better understanding of these plant functions. The present method would meet these needs, as shown in the results, in which Chl spatial distributions in each tree sample were estimated, and the heterogeneity of the Chl spatial distribution and the difference of the distributions between each sample are clearly observable. Chl vertical distributions have been estimated in previous studies, e.g., a study of the light absorption variation property for improving plant modeling [45] or the study of the differences in photosynthetic acclimation among different tree species or conditions [38,39,46–48]. A similar vertical variation of the Chl distribution at each height and for the sample could be efficiently estimated by the present method. Chl measurements in the previous studies were often conducted by labor-intensive destructive methods, which are often difficult to implement for obtaining data from the higher parts of the canopy. The present method is advantageous in terms of the data collection efficiency and ability to get information from previously inaccessible parts of the canopy. Autumnal leaf senescence changes the Chl contents within the leaves [40,49]. Since autumnal leaf senescence of sample A had already started, the Chl content distribution may reflect the differences in the progress of leaf senescence in each part of the canopy. The Chl vertical distributions of each orientation in the samples were different from each other. The light environment at each part of the canopy is

affected by the structure of the canopy itself and that of the surrounding trees, as reported in other studies, e.g., those of forest gap analysis by [50,51]. The differences in the quantity and direction of light incidence into the canopy in each orientation may result in different Chl distributions in each orientation. Besides the mean Chl values in each layer, the present method could provide the horizontal distribution of Chl values within the canopy (Figure 6). The full representation of the 3D distribution of Chl contents with enough spatial resolution in Figure 6 indicates that the sample trees distributed Chl unevenly and three-dimensionally, which illustrates the heterogeneous distribution of plant pigments. Although such information is essential to understand plant functions [45], data collection is typically difficult and detailed heterogeneous properties of the whole canopy up to the internal have not been estimated previously. The information of Chl distribution obtained by the present method can also be utilized to study the relationships between plant physiological processes, their structural properties, and environmental conditions in detail.

Novelty and usefulness of the present fusion method emerge through comparison between the previous methods and the present one. Some of the previous fusion systems between the lidar and 2D images were designed and assembled for a specific 2D camera and lidar, where the positional relation of the 2D camera and lidar was fixed by frames or other members [26,27]. In these systems, other 2D cameras and lidars are difficult to use at the same fixed arrangement and settings because the positional relation needs to be changed depending on the size or form of the camera and lidar, and according to image properties, such as resolution, view angle, and distortion. In the present method, the positions of the camera and lidar were adjusted in order to obtain similar projected images. Since the camera and lidar were not fixed to each other, the positions could be determined freely according to the target size or camera view angle. The 2D image registration technique of projective transformation also allowed flexible matching of images taken by different types of cameras and lidars. Such features of the present method illustrate its novelty and this method allows the fusion of images from different types of cameras and lidars; both multispectral and thermal cameras, which are completely different imaging devices, could be easily composited to obtain 3D lidar images using our method.

## 5. Conclusions

To obtain a 3D distribution of physiological and biochemical plant properties, we have proposed a fusion method between the 2D camera and 3D portable lidar images of plants. This method provides Chl spatial distributions of each tree sample, which includes the vertical and horizontal distributions, and those in each orientation. The resultant distributions show not only the average values in each layer or orientation but also horizontal distributions, which provide Chl contents of each part of the canopy. Such detailed 3D spatial information about plant biochemical properties cannot be obtained by conventional 2D-based methods. This method can be applied to different cameras in plant measurements and is particularly useful for studying plant physiological and biochemical properties. In addition to the study of plant properties, this method can also be used to investigate the relationships between the physiological, biochemical, and structural properties of plants, using various types of cameras and lidars.

**Author Contributions:** Conceptualization and methodology, F.H. and K.K; analysis, validation and visualization, S.U. and F.H, writing—original draft preparation, review and editing; F.H., supervision, funding acquisition, F.H.

**Funding:** This work was supported by JSPS KAKENHI Grant Numbers JP 17H03898.

**Conflicts of Interest:** The authors declare no conflict of interest.

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
