# Peer review of "Estimating 3D Chlorophyll Content Distribution of Trees Using an Image Fusion Method Between 2D Camera and 3D Portable Scanning Lidar"

_remotesensing, doi:10.3390/rs11182134_

Round 1

Reviewer 1 Report

This manuscript applied data fusion approach to 3D Lidar and 2D spectral images. The 3D point clouds are projected into the 2D image so as to add the chl content information which is converted from VIs into the SPAD values. The manuscript is concisely summarized and the proposed approach is new and of interest in the broad readers in the RS journal. Thus, the manuscript can be accepted after the minor modifications.

Comments and questions to the authors

I just wanted to ask the authors to clarify the effect of the LiDAR pulse footprints. As the longer the light path, the footprint becomes larger. Thus, the size of point clouds may depend on the observation-target distances. The authors did not show the absolute value of chl distribution. Chl content was represented by the SPAD value. It is known that SPAD and Chl have very good correlation. In the abstract and other appropriate places, it is better to mention that SPAD values should be converted by the SPAD vs Chi relationship that may be species dependent. In the methodology section, the derivation of canopy temperature by the thermal camera should be described more in detail (e.g. the assumption of emissivity of leaf and other materials, atmospheric effects are negligible or not). Line 93& 103: please indicate the exact date of the measurements. The conditions of leaves are very much different between beginning and end of October. Also, please describe the sky (cloudy) condition when measurements were performed.

Author Response

For reviewer 1
Please note that corrected parts based on your comments are highlighted using Track Changes function in the revised manuscript.

Comments

I just wanted to ask the authors to clarify the effect of the LiDAR pulse footprints. As the longer the light path, the footprint becomes larger. Thus, the size of point clouds may depend on the observation-target distances. The authors did not show the absolute value of chl distribution. Chl content was represented by the SPAD value. It is known that SPAD and Chl have very good correlation. In the abstract and other appropriate places, it is better to mention that SPAD values should be converted by the SPAD vs Chi relationship that may be species dependent. In the methodology section, the derivation of canopy temperature by the thermal camera should be described more in detail (e.g. the assumption of emissivity of leaf and other materials, atmospheric effects are negligible or not). Line 93& 103: please indicate the exact date of the measurements. The conditions of leaves are very much different between beginning and end of October. Also, please describe the sky (cloudy) condition when measurements were performed.

Response

As suggested, the length of the laser path determine the footprint size and horizontal resolution of the lidar image (attached to L109-110). In this study, SPAD values were used for the substitute of actual Chl contents but SPAD values can be converted into the actual Chl values using the correlation equation, as needed (added to L184). The spectral range and emissivity were attached to for more detailed specification(L201-203). The exact dates of the measurements were added (L94, 103). The sky conditions were attached (L94,104).

Reviewer 2 Report

Hi,

Overall, the paper has been presented well and sounds interesting for me. However, the statistical analysis are not strongly developed and need more in depth analysis. I havent seen any novelty discussion about the methodology and even though your work has good background, this section should be more highlighted.

Thanks

Author Response

For reviewer 2
Please note that corrected parts based on your comments are highlighted using Track Changes function in the revised manuscript. 

Comments

Overall, the paper has been presented well and sounds interesting for me. However, the statistical analysis are not strongly developed and need more in depth analysis. I havent seen any novelty discussion about the methodology and even though your work has good background, this section should be more highlighted.

Response

Thank you for your comments. The aim of this paper is to show the usefulness of the proposed 2D-3D fusion method and to demonstrate what kind of 3D structural data of trees can be acquired using this method. Thus, analysis of the obtained Chl distribution data was not strong in this paper. After the demonstration, detailed statistical analysis should be conducted to the data extracted from various plant species to investigate and understand the mechanism of plant functions. The novelty of the method is explained in L323-337, some descriptions were added (L323-324, L334) to highlight the contents. 

Reviewer 3 Report

This manuscript proposed a method to estimate 3D chlorophyll content distribution by registering 3d lidar points and 2d image pixels. The topic of this manuscript is interesting and potentially very important to the field. However, the manuscript is not well organized. The contribution of this paper is not clear. And the image fusion method doesn’t have enough novelty.

1. The key principle of the proposed image fusion method is to obtain similar projected images. Camera positions will have a significant impact on registration accuracy. No details are provided on how to mount the camera on the lines connecting the sample center and lidar positions.

2. Line 150-152: Please elaborate more about the 2d registration. How to choose corresponding points? Was it automatic or manual? How many pairs of corresponding points were chosen to get the projective transformation matrix? How to find projective transformation with corresponding points?

3. How many lidar images were used?

4. Line 170-174: More details should be provided on how the conversion equations were obtained. How many leaves are used? How to select these leaves? It seems like the relationships between VI and SPAD were different for different sample. How to determine which equation (e.g. linear, quadratic, exponential, etc...) to use?

5. Figure 6. Please make sure the colorbar labels are correct. Why the background cells have non-zero SPAD values?

Author Response

For reviewer 3
Please note that corrected parts based on your comments are highlighted using Track Changes function in the revised manuscript. 

Overall Comments

This manuscript proposed a method to estimate 3D chlorophyll content distribution by registering 3d lidar points and 2d image pixels. The topic of this manuscript is interesting and potentially very important to the field. However, the manuscript is not well organized. The contribution of this paper is not clear. And the image fusion method doesn’t have enough novelty.

Response

 Contribution of the paper is demonstration of an image fusion method that allows easy and accurate fusion of plant images between various 2D cameras and 3D lidars. Novelties of the method such as camera position settings and manner of 2D and 3D image registration are explained in the paper (L323-337), and highlighted. 

Minor comments

Comment 1

The key principle of the proposed image fusion method is to obtain similar projected images. Camera positions will have a significant impact on registration accuracy. No details are provided on how to mount the camera on the lines connecting the sample center and lidar positions.

Response1

In this method, the sample center, the lidar position and the center of the camera monitor were aligned on a straight line by viewing the camera monitor and moving camera body (added to L130-131).

Comment 2.

Line 150-152: Please elaborate more about the 2d registration. How to choose corresponding points? Was it automatic or manual? How many pairs of corresponding points were chosen to get the projective transformation matrix? How to find projective transformation with corresponding points?

Response2

Corresponding points were chosen manually by eye (L161-162). Five pairs were chosen for the projective transformation (L161). The transformation matrix was estimated by a least squares method based on the corresponding points(L164-165).

Comment3. How many lidar images were used?

Response3

Three images were used for each sample (added to L169 )

Comment4.

Line 170-174: More details should be provided on how the conversion equations were obtained. How many leaves are used? How to select these leaves? It seems like the relationships between VI and SPAD were different for different sample. How to determine which equation (e.g. linear, quadratic, exponential, etc...) to use?

Resonse4

Leaves were randomly selected from samples and they were marked by white markers. The numbers of the leaves were 20, 19 and 29 for sample A,B and C, respectively (L185-190). In the case of the sample A, exponential curve fitting was conducted due to NDVI saturation for SPAD values. Such phenomena was often observed in high Chl values. Linear fitting was conducted for the other samples due to no GNDVI saturation. (L193-195)

Comment 5.

Figure 6. Please make sure the colorbar labels are correct. Why the background cells have non-zero SPAD values?

Response5

Corrected (see Fig.6).

Round 2

Reviewer 3 Report

The authors followed the suggested changes. The manuscript has been improved.